# An Improved Indicator System for Evaluating the Progress of Sustainable Development Goals (SDGs) Sub-Target 9.1 in County Level

**Jiacheng Xu [1], Jianjun Bai [1,\*] and Jun Chen [2]**

[1]  School of Geography and Tourism, Shaanxi Normal University, Xi'an 713100, China
[2]  National Geomatics Center of China, Beijing 100830, China
\*  Correspondence: bjj@snnu.edu.cn

**Abstract:** In order to assess the progress of the SDG sub-target 9.1 at the county level, the SDG indicator 9.1.1 (rural access index) and 9.1.2 (passenger and freight volumes) were implemented in Deqing County, China to explore the fitness-for-purpose of these indicators for county level evaluations. It is found that the country-oriented indicator system has some localization problems and cannot fully reflect the connotation of the SDG sub-target 9.1 when used in the county level. An improved indicator system was built by modifying the SDG indicator 9.1.1 and adding three more indicators (namely the road density, accessibility, and total postal business). The analysis of the calculation process and results showed that the improved indicator system can solve the problems arising from the original SDG indicator when applied in the county level. The modified resident access index can eliminate the dependence of the original indicator 9.1.1 calculations on urban-rural boundary data, and takes into account the urban vulnerable groups such as urban villages residents. While the road density and accessibility can be used to measure the quantity, quality, and connectivity of the road and the reality of the residents to obtain the road, which enables the indicators to reflect the necessary details of the level of the transportation infrastructure construction. The total postal business can help the SDG indicator 9.1.2 reflect the relationship between the transportation infrastructure construction and the development of the economic and people's livelihood.

**Keywords:** SDGs; county level; improved indicator system; transportation infrastructure construction

---

## 1. Introduction

The 2030 Agenda for Sustainable Development has been adopted unanimously by 193 Member States at the United Nations Summit on Sustainable Development, in September 2015. The agenda covers 17 Sustainable Development Goals (SDGs), including those related to poverty, inequality, climate, environmental degradation, prosperity, and peace and justice [1]. The SDGs came into effect on 1 January 2016, and these new targets are applicable to all countries.

Since the establishment of SDGs, different researches on the entire target or some specific indicators from different aspects have been done by scholars. For example, Raszkowski and Bartniczak [2] research the implementation of the 2030 Agenda SDGs in Poland; the technical secretariat for the Statistical Coordination Group for the 2030 Agenda in Latin America and the Caribbean [3] has developed a regional indicator framework for SDGs based on the UN SDGs; Panda and Mohanty [4] discussed the progress of health-related sustainability indicators in Indian states; Li et al. [5] reviewed the progress of indicator 3.2.1 under-five mortality in 35 countries in Africa. Most of these studies focused on the application of indicators, such as using indicators to evaluate and monitor the level of sustainable development in some countries or regions. In addition, other parts of the studies focused

on the construction and improvement of the indicator system. SDGs are indeed a comparatively good indicator system to assess the sustainable development of countries around the world so far, while some studies have indicated that there are still some problems within it, some of them are listed as follows.

One important problem is that many scholars believe that the SDG indicators are far from perfect, the evaluation indicators set by the United Nations for each goal may not fully reflect the connotation of the target. For example, Giles-Corti et al. [6] believe that SDGs can't provide the upstream policies and interventions required to achieve the desirable health and sustainability outcomes specified in the SDGs, a more comprehensive set of indicators for cities could be developed; Guppy et al. [7] regard that there is a potential gap between the aspirations captured in core SDG 6 targets and the indicators that will be monitored, complementary indicators should be developed to narrow this gap; Unterhalter [8] believes that the development of the indicators for SDG 4 has resulted in metrics that miss many of the values of the targets, a deeper understanding and application of indicators should be done; the study by Brussel et al. [9] shows that the SDG indicator 11.2 supply oriented focus neglects the transport demand, oversimplifies the transport system and hides existing inequalities, and proposes two more reasonable alternative indicators.

The other problem of the SDG indicator system is that the designs concept and the data support of the SDG indicator system are country-oriented, and it is applicable to assess the progress toward the SDG targets for each country as a whole. However, an uneven development within a country often makes the country-oriented indicator calculation results be unrepresentative. Therefore, the SDG indicators often need to be applied to small internal areas in these countries. Thus, it is worthwhile to explore whether there will be problems when applying the SDG indicators to estimate small-scale areas (such as city or county). Until now, little research has been done in this area.

The transportation infrastructure plays an important role in sustainable development assessment. It has a bearing on agriculture, trade, education, medical care, etc., and has significant influences on the transportation costs, employment, accessibility of social services, and coordinated development between regions. It is, therefore, an important condition for achieving an inclusive and sustainable growth environment, and also an important factor restricting access to social and administrative services in the region [10–14]. In the SDGs, two indicators (indicator 9.1.1 and 9.1.2) in sub-target 9.1 focus on assessing the level of transportation infrastructure construction and its impacts on economic development and people's well-being. Unfortunately, little studies on these two indicators can be found in existing researches.

In this paper, the application of the SDG sub-target 9.1 in the county level region is mainly explored. The mentioned administrative region is the third level in China, usually under the jurisdiction of the prefecture-level (second-level administrative region), and a small part of it is directly under the jurisdiction of the provincial-level administrative region (the first-level administrative region). There are 2844 county-level administrative regions in China. In order to apply the SDG sub-target 9.1 in the county level, this paper in the second section takes Deqing County, Huzhou City, and Zhejiang Province in China as an example area to explore what, if any, are the problems and deficiencies of the SDG indicators 9.1.1 and 9.1.2 in measuring progress towards the goal 9.1 in a county level region. Section 3 presents some criteria for the indicators selection, the details of the various indicators that make up the improved indicator system, and the collection of data required for the indicators calculation. Section 4 lists some results computed from the improved indicator system and its advantages when applied to small scale areas compared with the original SDG indicators. Finally, we discuss the problems that exist in the improved indicator system when applied and the issues to be verified.

## 2. The Problem of the Original Indicators of SDG Sub-Target 9.1 When Applied in County Level

The official connotation description of sub-target 9.1 by the United Nations in the 2030 Agenda for Sustainable Development is: Develop quality, reliable, sustainable, and resilient infrastructure, including regional and transborder infrastructure, to support economic development and human

well-being, with a focus on affordable and equitable access for all [15], which is the aspiration captured in the SDG sub-target 9.1. It contains two specific indicators: "indicator 9.1.1—Proportion of the rural population who live within 2 km of an all-season road" (referred to as the rural access index—RAI) and "indicator 9.1.2—Passenger and freight volumes, by mode of transport "(referred to as passenger and freight volumes—PFV). Indicator 9.1.1 can usually be computed with Equation (1). The data required for indicator 9.1.2 are statistical data. Freight is calculated in tons. The cargo is counted according to the actual weight regardless of the length and the type of cargo. Passengers are counted by person, whatever the fare and the length of the itinerary, passengers are counted once per person.

$$R = RP_{LWR}/RP_{ALL}*100\% \qquad (1)$$

where R is the proportion of the rural population living within 2 km from a road that can travel all seasons to the total number of the rural population in the region, $RP_{LWR}$ is the rural population who live within 2 km from the all-season road, $RP_{ALL}$ is the total rural population in the region.

In order to find the problems that occur when using these two original SDG indicators in the county level, we calculated and analyzed the SDG indicators 9.1.1 and 9.1.2 using data from Deqing County, Huzhou City, and Zhejiang Province in China and compared it with the actual situation in the local area from the perspective of the SDG sub-target 9.1, and found that the SDG indicators 9.1.1 and 9.1.2 indeed fail to properly reflect the construction of the transportation infrastructure well when used in the county scale assessment. Basically, they are put in the following problems.

Firstly, the SDG indicators 9.1.1 and 9.1.2 cannot fully reflect the connotation of the SDG sub-target 9.1. Based on the official description of the SDG sub-target 9.1, we summarize the two requirements that a "qualified transportation infrastructure" should have, namely:

1. The requirements of the infrastructure itself. First of all, it should be able to withstand disasters and be sustainable; second, it should not only be able to be used in the region, but also capable to communicate with outside world, i.e., cross-border. In the end, it should be fair and accessible and affordable for everyone.
2. Utility requirements for the transportation infrastructure construction. It should be able to support economic development, protect the basic life of the users, and have the potential to improve the quality of life of the users.

For this purpose, the selected indicators should reflect at least the above two requirements to fully evaluate the sub-target 9.1. Unfortunately, the two SDG indicators cannot satisfy these requirements.

The SDG indicator 9.1.1 mainly focuses on whether the rural population can use the transportation infrastructure easily, which can reflect the fairness, but not reflect the quantity, quality, distribution connectivity or accessibility of roads. It should be noted that a road can usually only enter from the entrance, not any location on the road. For example, roads with guardrails or flyovers cannot directly be entered from a position other than the entrance, and the higher the level of the road, the more so. Therefore, there is an essential difference between "can find the road" and "can find the road entrance" within 2 km. It is evident that indicator 9.1.1 ignores this requirement. Overall, these limitations make it impossible to meet the first requirement.

Indicator 9.1.2 mainly measures the relationship between the transportation infrastructure and economic development, and hence, it does not reflect the impacts of the transportation infrastructure on improving people's well-being.

Moreover, in terms of the independent indicators, we find that there are still several problems with indicator 9.1.1.

1. Indicator 9.1.1 focuses on the accessibility of rural populations, and it does not realize that rural areas are not the only one that has poor accessibility in many regions (especially those in low- and middle-income countries). In order to achieve the goal of poverty alleviation, rural areas should be the focus for consideration, because the access gap is indeed widespread in rural areas,

and the construction of the transportation infrastructure is relatively less [10]. However, if the level of the transportation infrastructure construction of the whole region is determined by the accessibility of the rural population, it is implied that the level of the transportation infrastructure construction in non-rural areas (urban areas) is generally better than that in the rural areas. This seems undoubtful, but factually unreasonable. As shown in Figure 1, we can see that in the Xinshi Town, the sub-center city of Deqing County in the eastern part of Deqing (the red box), there is an area outside the buffer zone. We found in field investigations that the permanent residents in this area are not only the rural population, but also the urban population. Such a mixed rural and urban population area is not an exception in Deqing County, it is common in China and many other developing countries [16–18].

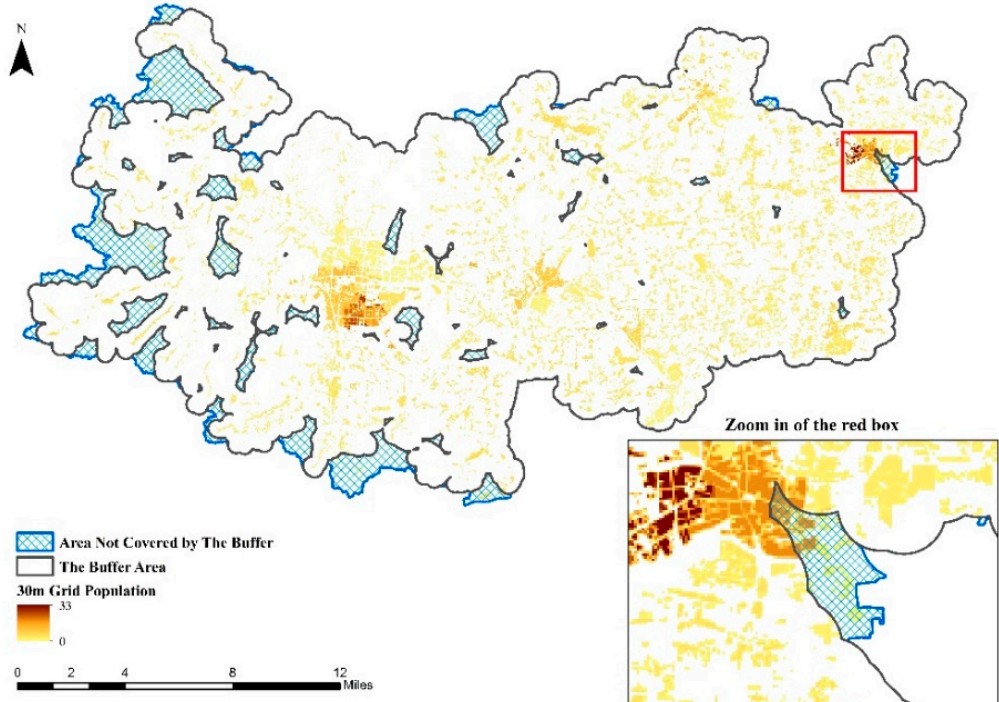

**Figure 1.** The 500 m buffer of the all-season roads in 2014. (The zoom in of the red box is a buffer-uncovered area in the Xinshi Town, where the urban and rural population live together). (The buffer is a service area for a geospatial target, specifically a polygon of a certain width built around points, lines, and polygons).

Many underdeveloped regions in the world (such as East Asia, most of West Asia and Central Asia) are facing enormous challenges in the process of urbanization. The low urbanization level and the poor capacity to provide the necessary and basic infrastructure for the increasing number of urban residents make these areas prone to "pseudo-urbanization" [19–21]. In China, many "urban villages" have emerged [22]. These residential areas lag behind the step of time development, lack of modern city management and have a low living standard in the process of rapid urban development [23,24]. Farmers in these areas have been converted to residents in the process of urbanization, but these "urban poor" still cannot get the proper transportation infrastructure services [25]. That means if we only concentrated on rural residents, it will be easy to draw conclusions that are inconsistent with the actual development in the region, which is unfavorable for sustainable development in this region. Therefore, the quality of life of "vulnerable groups" among urban residents should also be understood, in addition to paying attention to the convenience of rural residents in accessing the transportation infrastructure. Therefore, we propose a comprehensive evaluation carried out in the regional access, covering both the rural and urban populations [26–28] in this paper.

2. Indicator 9.1.1 has some difficulties in obtaining the data required for the calculation. In the calculation of the RAI indicator, the urban-rural boundary data is theoretically required to exclude the urban population from the calculation. However, in many counties, especially most of the underdeveloped regions, such data is generally lacking. In the case study of Deqing, the local management departments were also unable to provide us with good urban-rural boundary data.

According to the World Bank's recommendation, the result data of the Global Rural-urban Mapping Project (GRUMP), which included the "urban-wide data" can be used to extract rural areas [10] and to calculate this indicator. However, this data is estimated to be derived from long-term observations of nighttime light data to identify areas that appeared to be urbanized. The resolution of this data is 30 arc- seconds (approximately 1 km) [29]. When used for smaller scale studies, it may cause a large error. When conducting research in county level areas, wherever conditions permit, a household survey can be used. However, the data collected by this method is sometimes incomplete, and the cost is often expensive, which makes the indicators unsustainable. Therefore, the household survey is not a good way to solve the mentioned problems. Meanwhile, it is difficult to require the management component to produce this data in low- and middle-income countries. Take China as an example. Most cities in China are not geographically urbanized areas, but an administrative division [30]. Therefore, the area of the city does not geographically reflect the area of the city, i.e., the scope of urbanization [31]. Thus, there is no clear boundary between urban and rural areas due to rapid urbanization. In many areas, there are "urban villages" and "development areas". Moreover, these areas tend to have a large floating population, which in turn makes the boundaries between the rural and the urban population further blurred. In a word, the lack of urban boundary line data makes the RAI indicator difficult to be calculated.

Based on the above discussions, an improved indicator system that is suitable for the county level research and can comprehensively evaluate the level of the regional transportation infrastructure construction is established to solve these localization issues of the two indicators of SDG sub-target 9.1 in the county level application.

The purpose of the next research in this paper is mainly involved in two aspects:

1. Eliminate the defects of the SDG sub-target 9.1, and establish an improved indicator system in accordance with the sub-target 9.1 requirements.
2. Analyze the calculation results from the improved indicator system in Deqing County, and measure the advantage or disadvantage of the improved indicator system when applied in the county level.

## 3. Materials and Methods

### 3.1. Research Area

Deqing County is located in the north of Zhejiang Province, covering an area of 937.92 km$^2$. It has a jurisdiction over eight towns, four streets, and a registered population of 430,000. In 2017, the regional GDP was 47.02 billion yuan. The main road network consists of national highways and highways running through the county, appearing as a distribution of "three verticals and one horizontal" in space, as shown in Figure 2.

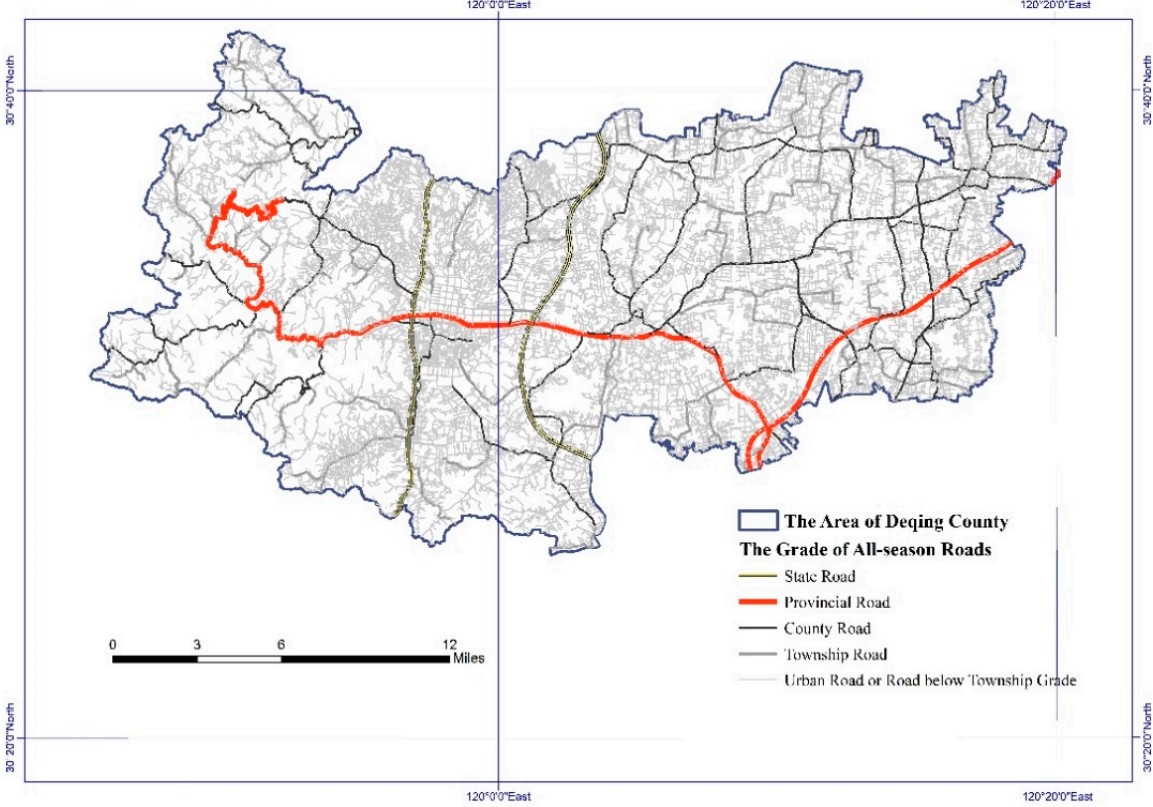

**Figure 2.** The transportation network in Deqing County in 2017.

### 3.2. Methodology

For the purpose of the problems presented in Section 2 of this paper, we examined the indicators that are commonly used to evaluate the level of the transportation infrastructure construction [32–35] to develop the improved indicator system. We consider a number of criteria to be important when selecting a new indicator:

1.   Data should be easily obtained or generated by less difficult methods;
2.   It should be applicable to county level applications;
3.   It should be easy to explain and can clearly convey its calculation results.

Finally, we selected five indicators to build the improved indicator system, as shown in Table 1.

**Table 1.** The list of the improved indicator system.

| Indicator Index | Sub-Indicator Index | Indicator Name |
|---|---|---|
| II1 | II1.a | Road Density (RD) |
| | II1.b | Resident Access Index (RAI) |
| | II1.c | Accessibility (Average Travel Time- ATM) |
| II2 | II2.a | Passenger and freight volumes, by mode of transport (PFV) |
| | II2.b | Total Postal Business (TPB) |

We divided the improved indicator system into two parts, numbered by II1 (Improved Indicator 1) and II2 (Improved Indicator 2), respectively. Indicator II1 is mainly used to evaluate the construction level of the transportation infrastructure itself. There are three sub-indicators, which are identified by a, b, and c, reflecting the quantity and accessibility of the transportation infrastructure. Sub-indicator II1.a RD is used to evaluate the quantity and spatial distribution of roads in the study area to show the most basic aspects of the transportation infrastructure construction. Sub-indicator II1.b RAI is

used to measure whether residents can obtain a certain quality standard within a certain distance. The sub-indicator II1.c ATM is used to judge the connectivity and quality of the road by calculating the average travel time (the higher the road quality, the stronger the traffic capacity, the less the average travel time). Combining the calculated results of these three indicators can provide a more comprehensive insight of the construction level of the transportation infrastructure in the study area. Indicator II2 is used to assess the relationship between the transport infrastructure, economic development and people's livelihood. It has two sub-indicators, which are identified by a and b, to evaluate the service level of the transportation infrastructure for economic development and people's livelihood together.

### 3.2.1. Sub-Indicator II1.a Road Density

This indicator is used to evaluate the level of the transport infrastructure development within the region from the "quantity" perspective, including RD and weighted road density. RD is the proportion of the total road length to the area of the region. It is one of the basic attributes of road traffic networks and a common indicator for evaluating the merits of a regional traffic network [36,37]. The calculated results of the road network density have an overarching significance for improving the adaptability of the transportation and economic sustainable development and the rational formulation of road network planning [38,39]. By using this indicator, we can analyze the spatial distribution characteristics, fairness, and rationality of the regional transportation infrastructure construction.

RD is calculated as follows: First, create a $100 \times 100$m grid covering the study area (the grid size is usually determined by the size of the study area). Then, count the total length of the road within each grid. Finally, use Equation (2) to get the road density.

$$RD = TLR/A \qquad (2)$$

where RD is the road density in km/km$^2$, TLR is the total road mileage in km, and A is the regional area which is 10,000 m$^2$ in this paper.

The calculation of the weighted road density first counts the road lengths in the grid according to the different road levels, then multiplies them by their respective weights, and finally sums them up. Regarding the setting of the road weights of different levels, the method of expert scoring is usually used to obtain the proportion of the road-level traffic capacity. The road weight values in this paper are determined by referring to other studies [40,41], as shown in Table 2.

**Table 2.** The weight of each road level when calculating the weighted road density.

| Road Grade | State Road | Provincial Road | County Road | Township Road | Urban Road or Road below Township Grade |
|---|---|---|---|---|---|
| Code | G | S | X | Y | Q |
| Weight | 0.4 | 0.21 | 0.18 | 0.13 | 0.08 |

### 3.2.2. Sub-Indicator II1.b Resident Access Index

The resident access index is the proportion of the residents who live within 2 km from an all-season road. There is a common understanding that the 2 km threshold is a reasonable extent for people's normal economic and social purposes [10]. The all-season road refers to the road that can be used throughout the year by the universal means of transportation in the region, and should be resilience to natural disasters (such as heavy rain) [42]. This indicator has evolved from the SDG indicator 9.1.1. For the reasons that China's urban-rural boundary is unknown and the level of urbanization is low, we have revised its calculation target to take into account urban areas. In this way, it also eliminates the impacts of data difficulty on the indicators calculations. This indicator is simple in definition and has a high relevancy with the operations of the regional road sector. It can be used not only to measure the level of road construction, but also in a broader development context, such as poverty alleviation [10].

The calculation method is as follows: First, using the all-season roads data to produce a 2 km buffer area. Then, overlaying the population density data and the buffer range data to calculate the sum of the populations falling within the buffer zone, i.e., the population who lives within 2 km from the all-season roads. Next, calculate the proportion of the population within the buffer area to the total population of the county, i.e., the result required by the indicator. It is also possible to separately calculate the urban population and the rural population, and calculate the coverage ratio of the two types of population. Finally, the population of the area not covered by the buffer can be counted according to the urban population and the rural population, and the proportion can be calculated. The number of urban and rural populations in areas not covered by the buffer can be separately counted if conditions permit.

### 3.2.3. Sub-Indicator II1.c Accessibility (Average Travel Time)

This indicator is used to reflect the connectivity of the road and to calculate the transit time cost to reflect the road quality. The SDG indicator 9.1.1 only evaluates whether roads are available and does not consider the connectivity of these roads. In reality, there may be some roads whose construction quality satisfies certain standards, but is disconnected to the regional transportation network. The RAI indicator of residents living around these "isolated" roads may be 100%, while the actual transportation of these populations may not be very well. Sub-indicator II1.c can remove the impact of these disconnected roads on the calculation results, and enable the results closer to the actual life of the residents. Moreover, the road network model is often constructed to simulate the actual road traffic when calculating the average travel time. As mentioned in the second section above, the SDG indicator 9.1.1 cannot be used to calculate whether the roads searched by residents within 2 km can be directly entered. Sub-indicator II1.c can solve this problem well because of the limitations of nodes such as road entrances and exits. On the other hand, the average travel time can be used to measure the regional traffic capacity, which also helps to analyze the spatial distribution of the road infrastructure construction in the study area. Compared with the first two indicators, sub-indicator II1.3 focuses on evaluating the quality of the transportation infrastructure rather than the quantity, which is a higher-level requirement for the transportation infrastructure construction [43,44].

This indicator is calculated as follows: First, build a road network dataset and use Equation (3) to compute the "travel time cost" for each road [45]. Second, divide the area into a regular grid and extract the center point of the grid. Third, calculate the time cost of the feature points (usually choose administrative center, economic center, etc.) in the area to the center point of each grid, and then average the average travel time of each grid point to each feature point of the area. Finally, a spatial display can be performed.

$$Cost = L/V \tag{3}$$

where Cost is the time cost when passing each road in a minute; $L$ is the length in km of each road; $V$ is the limited speed in km/h of each road, which should refer to the road construction standard documents of the area. This paper determines the data by referring to the "China National Road Construction Standard".

### 3.2.4. Sub-Indicator II2.a Passenger and Freight Volumes, by Mode of Transport

This indicator has clear definitions, statistical methods, and data sources in the original SDGs metadata, it will not go into details in this paper. For more information, the United Nations indicator metadata document can be viewed [46]. This indicator is used to assess the level of the regional transport infrastructure supporting economic development (cargo volume and passenger traffic) and basic livelihood (passenger traffic).

In terms of statistical data, the passenger and cargo traffic is usually counted at the national level according to different modes of transportation such as air transport, highway transport, road transport, rail transport, water transport, and intermodal transport, etc. At the county level, some

modes of transportation do not exist. Moreover, the source of statistical data has changed in county level studies. Data from international statistical offices are often used when evaluated in the national level. The International Civil Aviation Organization provides data on the transport (passenger and freight traffic) of its member countries in relation to air transport, and the International Transport Forum collects the annual transport (railway and highway) statistics for its member states. When evaluated in the county level, it has to be obtained from the regional management departments.

In the study of Deqing, we only collected the passenger and freight volume data of the road transport and water transport to adapt the local reality.

### 3.2.5. Sub-Indicator II2.b Total Postal Business

This indicator is used to evaluate whether the construction of the transportation infrastructure has a positive impact on the improvement of people's well-being [47]. The total postal business refers to the monetary performance of the total number of services provided by the postal sector to the society. The volume of freight is mainly related to economic development, while the postal business mainly meets the daily life and material needs of people. For the propose of "Logistics Performance Index", the World Bank defines the quality of the trade and transportation-related infrastructure as one of the important factors affecting the efficiency of trade logistics. Therefore, we can inference from the data changes in the total amount of postal services whether the level of the transportation infrastructure construction can meet the material needs of local residents [48–50] based on such correlations.

This indicator can usually be obtained directly from the regional management sectors. It also can be calculated using Equation (4).

$$TPB = (PBV * CP) + OPI \tag{4}$$

where TPB is the total postal business in yuan, PBV is the various postal business volume in yuan, CP is the constant price, which makes the amount of income in different years able to be compared and OPI is other postal income in yuan.

### 3.3. Data Collection

The road data, population distribution data, and passenger and freight volume statistics provided by the National Bureau of Statistics and the surveying and mapping departments can be used when calculating these indicators at the national level. Due to the fact that the standards for data collection may vary from country to country, using open source data from international organizations may make the application and promotion of the indicators more sustainable and the calculated results of different countries more comparable. For example, road data, population data, and country border data can be assessed from the OpenStreetMap, Worldpop project and GRUMP project, separately. The International Civil Aviation Organization provides its member countries with traffic data related to the air transport (e.g., passenger and freight volume). The International Transport Forum collects statistics on transportation (rail and road) from its member countries each year.

When calculating this indicator worldwide or over a large scale, the use of open access data sources will make the application and promotion of the indicators more sustainable. While doing calculations on the county level, we recommend obtaining data from the local management departments. There are several advantages of it:

1.  This will enable local management departments to fully understand the local developments during data collection, to identify problems timely, and to plan for further development;
2.  This also encourages managements to use the indicator calculations in its daily works to make indicators more sustainable [10];
3.  The data from local management departments is more credible than the open access data, and has local characteristics, which is conducive to calculating the results of the indicators that are more in line with the "local conditions".

Data from the higher-level management can be considered if it is unable to obtain it from local management departments. Data from the international cooperation departments or open access platforms can also be used.

Table 3 is the main data and their sources used in this paper.

**Table 3.** The main data and the data sources used in the Deqing case study.

| Number | Data | Data Source |
|--------|------|-------------|
| 1 | Road | The road data comes from the Deqing County Transportation Management Department. It is a vector data, which contains the level of the road. |
| 2 | Population | It is calculated. The residential coverage of all residents will be extracted from the surface coverage data obtained from the Deqing County Geographic Information Center, and then the 30 m grid will be used to divide the residential area. Finally, the census obtained from the Deqing County Statistics Bureau will be based on the housing level in the land use data. The data is distributed to the surface coverage data according to the location. In this way, the 30 m grid population density data can be generated. |
| 3 | Passenger and freight volumes | These data were obtained from the Deqing County Statistical Yearbook published by the Deqing County Bureau of Statistics. |
| 4 | Total postal business | |

## 4. Results

### 4.1. Calculated Results from the Improved Indicator System

#### 4.1.1. Road Density

As it can be seen from Figure 3, the spatial distribution of road density in Deqing County is relatively balanced. The average road density is 1.24 km/km$^2$, and the maximum value is 1.69 km/km$^2$. The higher road density is distributed mainly in the center of each town and near the built-up area. The road density near the high-level roads is generally higher. The minimum average road density and the standard deviation minimum are in Moganshan Town, which are 0.83 km/km$^2$ and 12.3, respectively. It indicates that the road density in Moganshan Town is generally low. The correlation coefficient between the population of each town and the road density is −0.09, indicating that the correlation between the two is not significant. As it can be seen from Figure 4, the maximum road density is distributed along the two highways. This shows that most of the roads in Deqing County are low-level roads. In short, the average road density and the average weighted road density of Fuxi Street are higher than those of other towns, and Moganshan Town for its mountainous area is ranked last. The spatial structure of the road density in Deqing County is a multi-center distribution of high-level roads. It can be seen from Figure 5 that the important public service facilities in Deqing County, such as schools and hospitals, have higher road densities and higher accessibility.

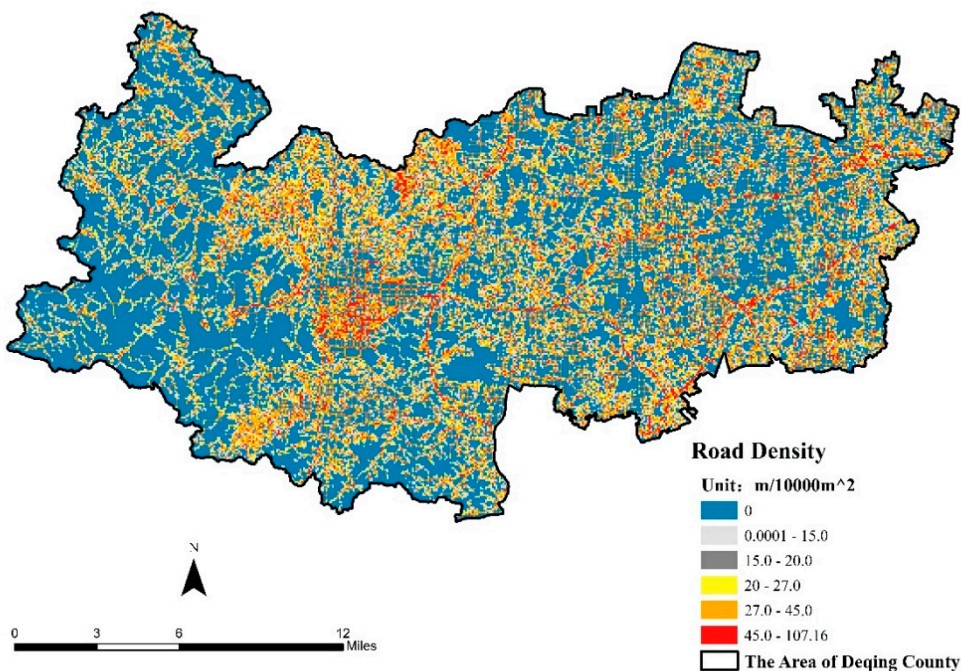

**Figure 3.** The spatial distribution of the road density in Deqing County in 2017.

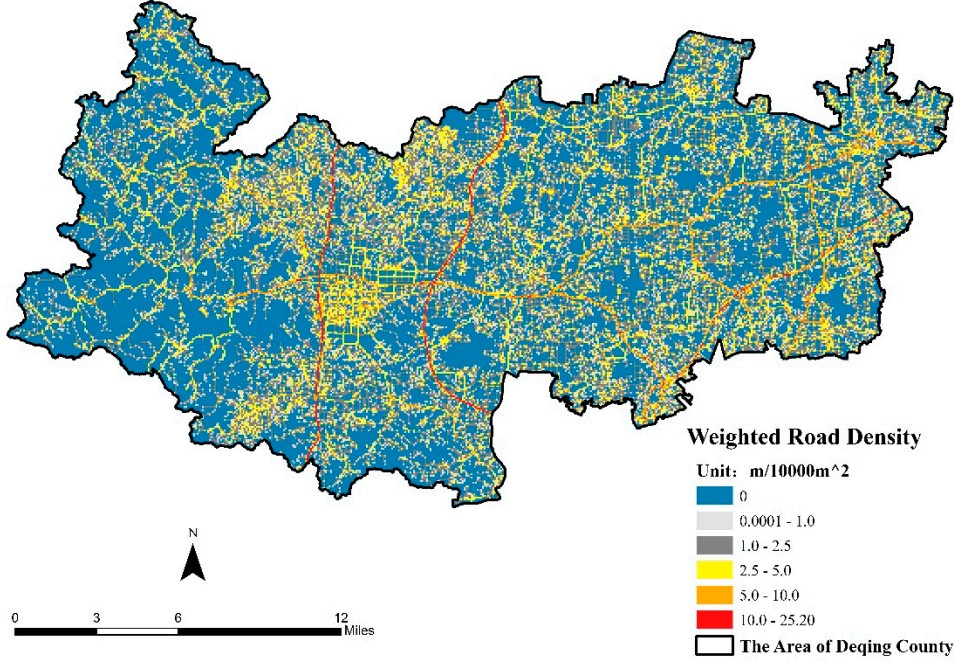

**Figure 4.** The spatial distribution of the weighted road density in Deqing County in 2017.

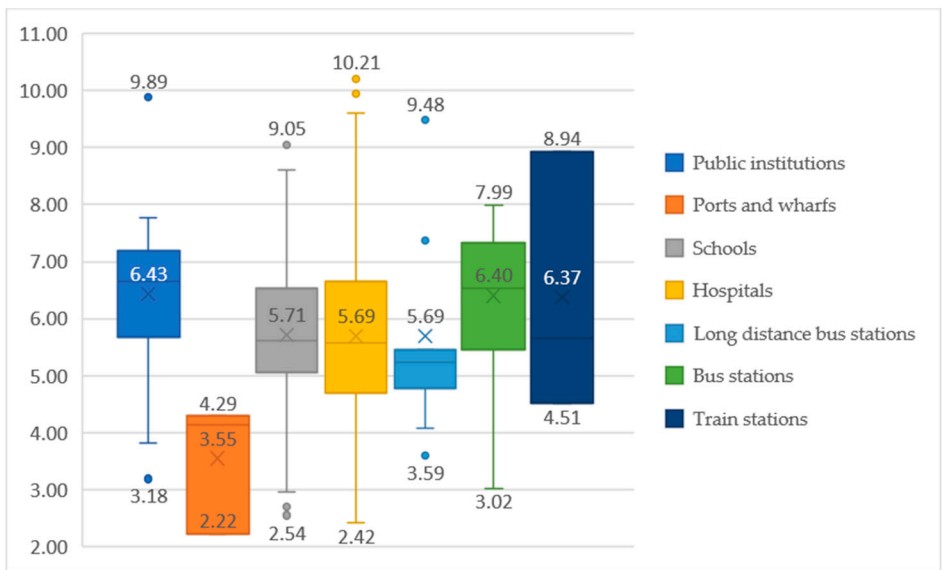

**Figure 5.** The road density at the basic public service facilities in Deqing County. (The numbers in each box are the maximum, average, and minimum density in km/km$^2$ from top to bottom.).

### 4.1.2. Resident Access Index

As shown in Figure 6, the RAI in Deqing County has already reached 100% in 2014. To explore in more details, we upgraded the threshold from 2 km to 500 m. The 500 m buffer zone of the all-season road in 2014 and 2017 are shown in Figure 7. In 2014, the area which was not covered by the 500 m buffer zone of the all-season road was 26.6 km$^2$, 2.83% of the total area of the county. Among them, the Xin'an Town, and Yuyue Town have reached full coverage. The uncovered proportions of towns such as the Moganshan Town, Wuyang Street, and Wukang Street are relatively high, at 7.77%, 5.69%, and 5.10%, respectively. In 2017, four towns and streets including the Fuxi Street, Zhongguan Town, Leidian Town, and Qianyuan Town also reached full coverage. The uncovered area of the county has been reduced to 4.34 km$^2$, of which the Moganshan Town reached 43.94%. Most of the area in the Moganshan Town is mountainous, and the road is relatively sparse than those in the eastern Deqing County. However, the Moganshan Town has a small population distribution and is mainly lived along the road. Although there are areas not covered by road buffers, the residential areas have been covered by buffers (in 2017).

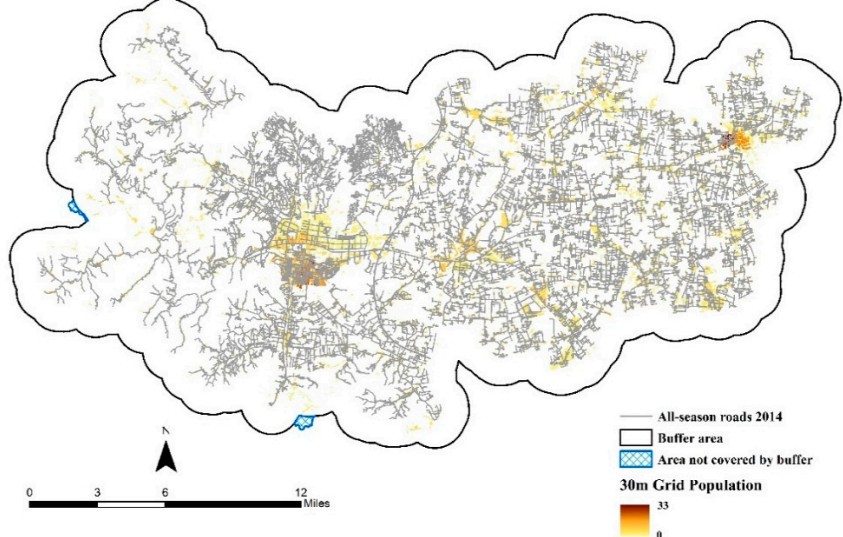

**Figure 6.** The 2km buffer of all-season roads in Deqing County in 2014.

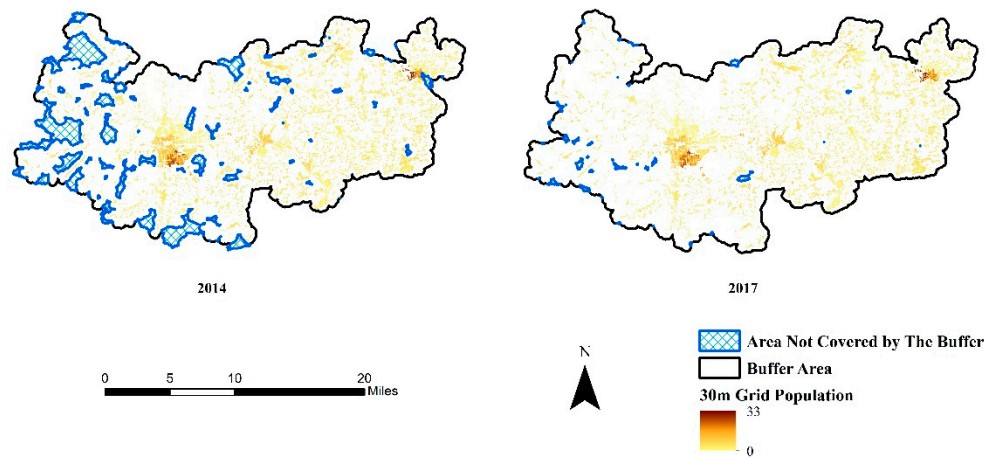

**Figure 7.** The 500 m buffer of the all-season roads in Deqing County in 2014 and 2017.

### 4.1.3. Accessibility (Average Travel Time)

As it can be seen from Figure 8, the northwestern part of the Moganshan Town in 2014 is less accessible than the others, and the fragment areas with a sharp drop in accessibility are scattered throughout many towns for two reasons: First, the extent and quantity of the road are insufficient. Second, there are roads but not connected to the road network in the county. This makes the region to be in an "isolated" state. By 2017, the traffic accessibility in Deqing County has been significantly improved. The change in accessibility is relatively uniform, and the degree of fragmentation is small, indicating that the area in the "isolated" state is significantly reduced, and the road connectivity is greatly improved. As it can be seen from Figure 9 and Table 4, the average travel time has increased significantly. The average of the county's ATM dropped from 31'42" (which means 31 min and 42 s) to 26'30". Especially in the northwestern part of the Moganshan Town and most of the Yuyue Town, the average travel time decreased by more than 15 min, with a maximum value of 44'48". As it can be seen from Figure 10, the proportion of the area of the county's average travel time within half an hour increased from 53.79% to 76.09%, while the proportion of the area larger than one hour decreased from 3.31% to 0.07%, showing that in addition to connectivity, the number and quality of roads have also been improved.

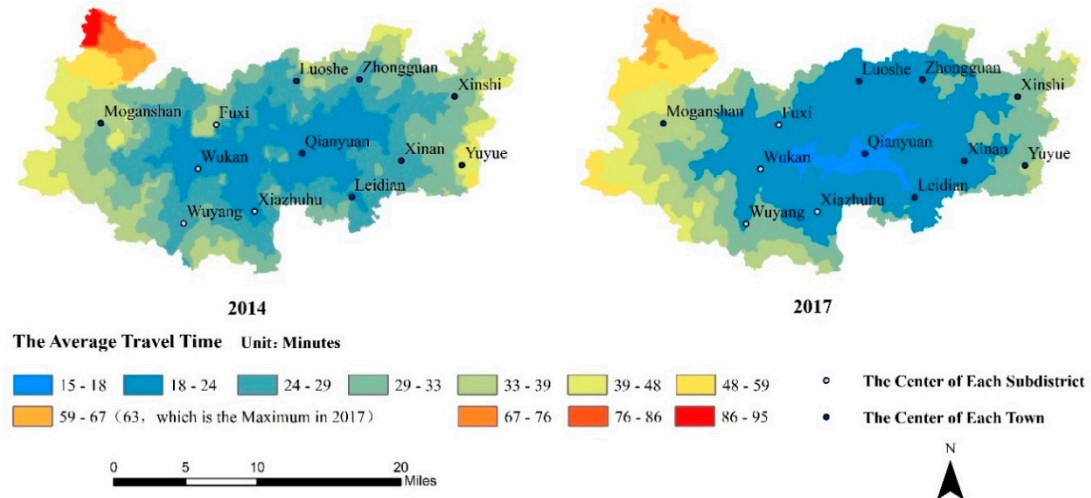

**Figure 8.** The spatial distribution of accessibility (average travel time) in Deqing County in 2014 and 2017.

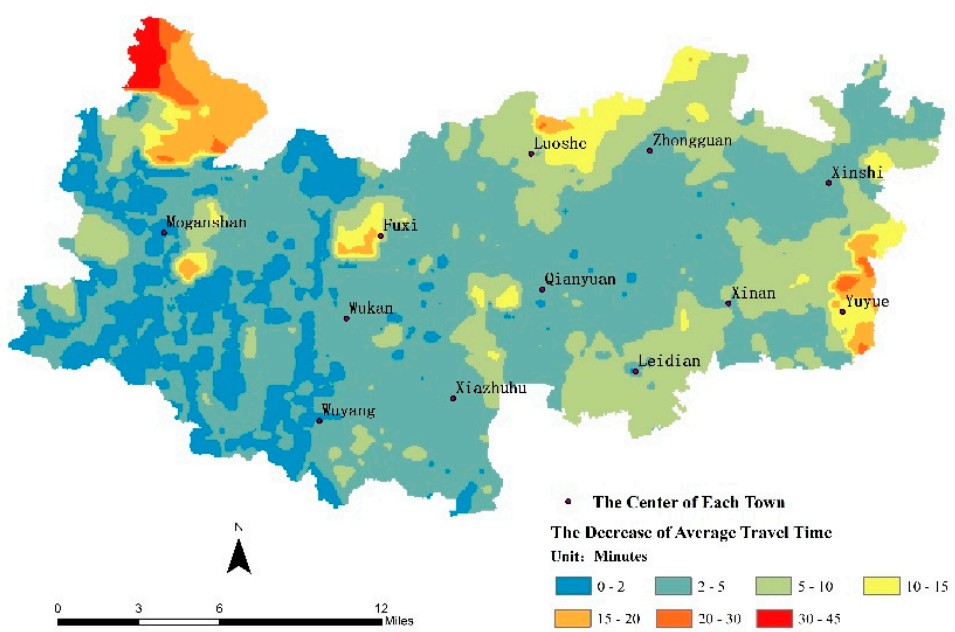

**Figure 9.** The spatial distribution of the progress of accessibility (also the decease of the average travel time) in Deqing County from 2014 to 2017.

**Table 4.** The average travel time in Deqing County in 2014 and 2017.

| Average Travel Time | 2014 | 2017 | Reduced Value |
|---|---|---|---|
| Min | 18'06" | 1'42" | 2'24" |
| Max | 94'47" | 62'58" | 31'49" |
| Average | 31'42" | 26'31" | 5'11" |
| Standard deviation | 626.31 | 444.56 | 181.75 |

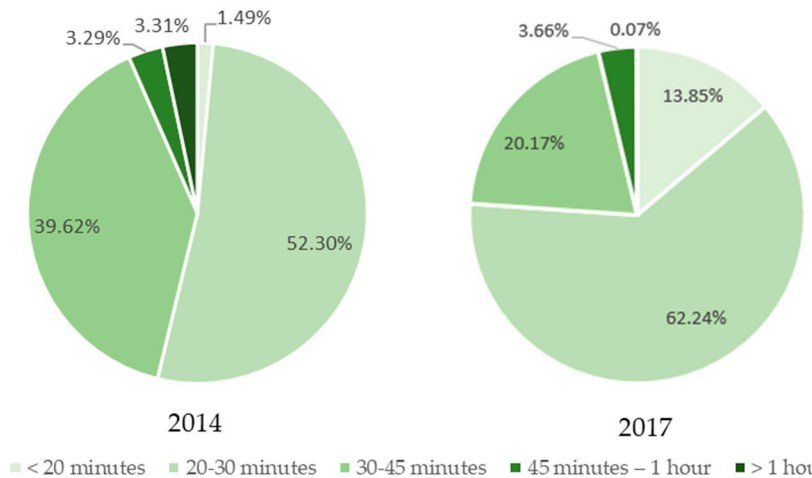

**Figure 10.** The statistics of the average travel time in Deqing County in 2014 and 2017.

The accessibility in Deqing County is characterized by gradual weakening from the center to the edge, and the change of accessibility is basically consistent with the extension direction of the "three vertical and one horizontal" traffic network. In 2014, the area with the highest accessibility was the Qianyuan Town, with an average of 23'18". The poor area was the Moganshan Town with an average of 44'30". The average travel time in the county is mostly in 18–39 min, with a maximum of 94'48". In 2017, the average travel time in the county has shortened significantly, and the maximum weighted average travel time has reduced to 63'0", lessened by 31'48". The significant drop in the standard

deviation indicates that the county's accessibility differences have declined and the accessibility became more balanced.

### 4.1.4. Passenger and Freight Volumes, by Mode of Transport

Figures 11 and 12 show the passenger and freight volumes in Deqing County over the years. It can be seen from these figures that the passenger and freight volumes in Deqing County have a downward trend in recent years. The road traffic volume and passenger turnover in Deqing County have started to fall from the peak in 2012 and since 2013 have decreased yearly. The total passenger volume (road and water transport) in Deqing County was 9.67 million in 2016, a decrease of 32.09% compared with that in 2012, and the road passenger volume decreased by 33.24%. In 2016, the passenger turnover was 249.75 million kilometers, a decrease of 42.3% compared with that in 2012; the freight volume (road and water transport) was 13.26 million tons, declined 1.72 million tons since 2015. Among them, the road freight volume decreased by a large margin. According to the records, the highest decline is in 2016, reaching 17.20%. Meanwhile, the freight turnover has also declined, but has rebounded slightly in recent years. The freight turnover totaled 1.21 billion ton-kilometers in 2016, a decrease of 38.58% compared with that in 2012, while it was up 10% from 2014 (the lowest level in recent years). Among them, the turnover of road cargo has rebounded rapidly and has reached the average level in recent years. The freight turnover has increased slightly in recent years, it still cannot reach the level before 2013.

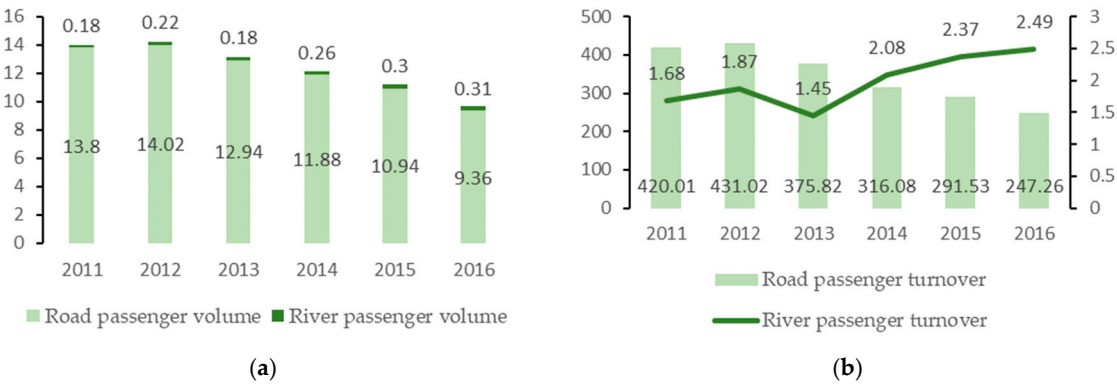

(a)  (b)

**Figure 11.** (**a**) The yearly passenger volume in Deqing County (Unit: Million); (**b**) The yearly passenger turnover in Deqing County (Unit: Million people kilometers).

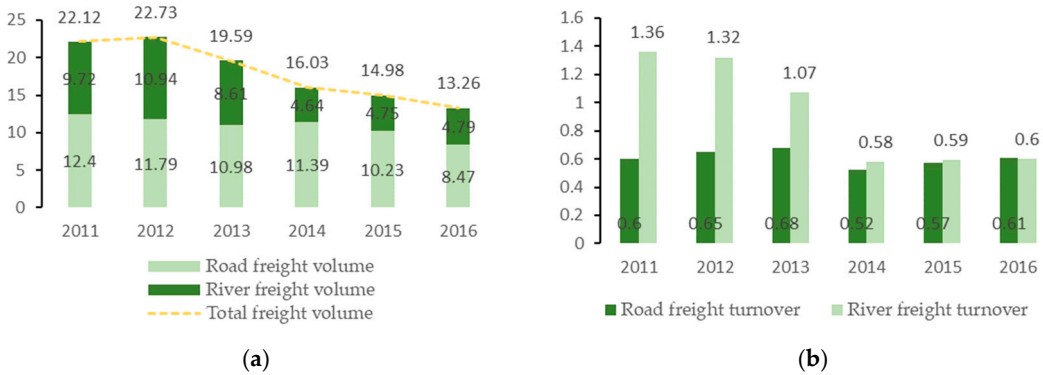

(a)  (b)

**Figure 12.** (**a**) The yearly freight volume in Deqing County (Unit: Million tons); (**b**) The yearly freight turnover in Deqing County (Unit: Billion tons kilometers).

### 4.1.5. Total Postal Business

As it can be seen from Figure 13, both the total postal business and the growth rate in Deqing County has been rising in recent years. In 2016, the total postal business increased by 32.89%, and the

number of express deliveries increased by about four times compared with that in 2011. This indicates that: First, the rapid increase in the express delivery demonstrated that remarkable achievements have been realized thanks to the construction of transportation infrastructure supporting its development and the blank road area was further filled, the road quality was further improved to make the working conditions of the express delivery improve; second, the increase in the total amount of postal services also shows the improvement in people's quality of life and level of consumption. The regional economic cannot get growth without the support of a good transportation infrastructure, through which the quality of people's life improves. Hence, through the improvement of the total postal service in Deqing County, it can be reasonably speculated that the transportation infrastructure has played a positive role in improving people's lives and well-being.

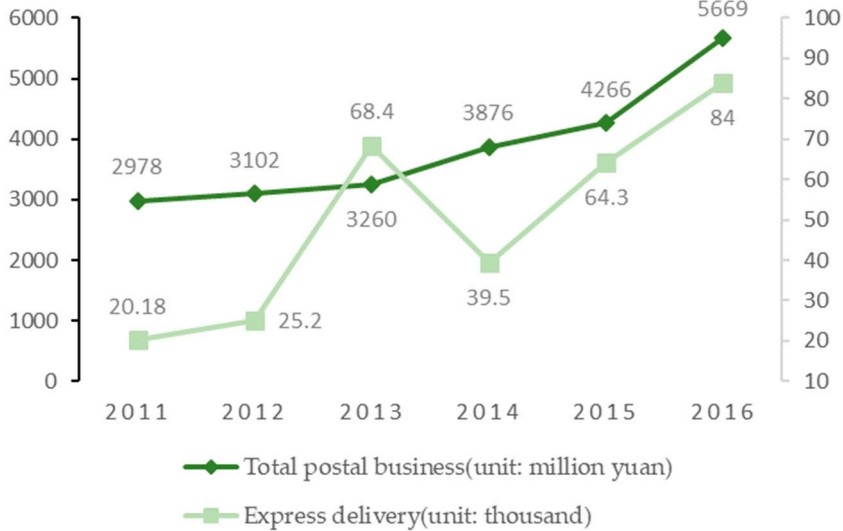

**Figure 13.** The yearly total postal business in Deqing county.

*4.2. The Advantages of the Improved Indicator System*

As for specific indicators of the improved indicator system, sub-indicator II1.a can assess the development level of the road in the region by the road density. Sub-indicator II1.b can basically determine the distance the residents live away from the road. Sub-indicator II1.c can determine the convenience of residents' travel according to the connectivity of the road. The three indicators can be used to obtain the construction level, availability and convenience of the transportation infrastructure in the area. Through sub-indicators II2.a and II2.b, the impacts of the transportation infrastructure on the economic construction and people's lives is gained. By comparing the calculation results of the SDG sub-target 9.1 and the improved indicator system as shown in Table 5, we can find that the improved indicator system can solve the problems mentioned in the second section above.

**Table 5.** The comparison of the original SDG indicators and the improved indicator system.

| Problem | Dependence on Urban Rural Boundary Data During Calculation | Take into Account Urban and Rural Population | Show more Details and Spatial Distribution of Road Construction | Reflect the Impact of Transportation Infrastructure on Economic Development and People's Lives |
|---|---|---|---|---|
| SDG indicator 9.1.1 and 9.1.2 | More | No | No | Incomplete |
| Improved indicator system | Less | Yes | Yes | More complete |

First, the improved indicator system provides a new insight for the calculation of the SDG indicator 9.1.1, taking into account the evaluation of the transportation infrastructure construction in urban areas. Theoretically, the RAI indicator considers both rural residents and urban residents, which makes "urban poor" such as urban floating population and urban village residents no longer be regarded as groups that have obtained good transportation infrastructure services. In this way, the urban population that was not covered by the buffer zone in the Xinshi Town in Deqing County in 2014 could be captured. Meanwhile, such calculations no longer have a strong dependence on the urban-rural boundary data, it thus increases the availability of indicators for county level applications. Once the calculation results are obtained, household surveys can be carried out in areas not covered by the 2 km road buffer when necessary, which can reduce the application cost of the indicators.

Second, the improved indicator system can comprehensively assess the construction of the transportation infrastructure in the study area, and the three new indicators added enable more details to be discovered. From the calculation of ATM and RD, a lot of information, such as the spatial imbalance, the concentration or lack of road construction, spatial distribution of different quality roads, spatial distribution of travel time within the county, and the spatial relationship between road distribution and population density, etc., can be derived to provide solutions and guidelines for future transportation infrastructure construction. For example, in the Deqing study case, we found that most areas in the Moganshan town and urban fringe areas have relatively poor accessibility. The road density in Deqing County has an aggregation phenomenon with an imbalanced spatial distribution, and the Moganshan Town is still the main area with a low value. These problems are hidden by the calculation result of the SDG indicator 9.1.1 (100% of the RAI indicator).

Third, the new indicator TPB added to evaluate the link between the transportation infrastructure and economic development and people's well-being, enables the mentioned improved indicator system to make an overall evaluation.

In short, we can find from the above comparisons that the improved indicator system is more suitable for the county level assessment, especially for low- and middle-income countries with rapid urbanization like China. Moreover, it can evaluate the level of the transportation infrastructure construction more comprehensively.

## 5. Discussions

In its application, we also found some problems.

1.  The SDG indicator 9.1.1 only requires roads to be accessible in all seasons, that is, once the roads meet this condition, they can be included in the calculation data. As for our view, besides this condition, the connectivity of the road should also be considered as a new condition for the reasons that even users can find qualified roads in the search range of 2 km, these roads don't have enough significance to promote the regional economic development and to meet the needs of residents' livelihood if they can't communicate with the outside world, or if there are sections with poor traffic capacity. In 2014, there were a large number of up-to-standard roads in the northern part of the Moganshan Town that were not connected to the county's transportation network, while they can provide little actual help when locals go to other towns and streets. Therefore, when calculating the RAI, the quality of the transport infrastructure services available is very different between the residents near these roads and those near roads with better connectivity. However, the calculation of the SDG indicator 9.1.1 cannot show this. As shown in Figure 14, whether or not to delete the isolated roads has an impact on the indicator calculation results.

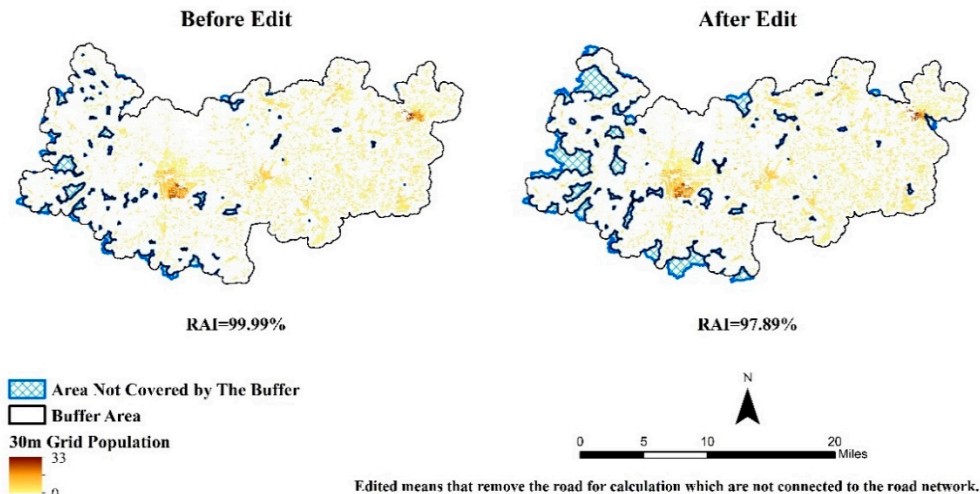

**Figure 14.** The comparison of the 500 m buffer of the all-season road in Deqing county in 2014 before and after editing the isolated road.

In the Deqing study area, when we calculated the new sub-indicator II1.b, the road data used had already deleted all the suspended roads that were not connected to the road network. We decided to calculate this indicator with more stringent requirements. However, whether this is reasonable or not is a matter of opinion. If the study area is a region with slow economic development, a suspended road may facilitate the residents in the area. Therefore, the requirements for the regional transportation infrastructure construction in underdeveloped areas may first be "whether there has road that meets the quality requirements" rather than "whether the connectivity of these road is good". Hence, if a poor result is obtained by rashly deleting the unconnected road, the recommendations for the regional development will instead confuse the primary and secondary. After all, areas without roads should be given more attention than those with poorly connected roads. As a result, it is difficult to say how to deal with the relationship between the availability and connectivity of the road.

2. The relationship between the passenger and freight volumes and transportation infrastructure construction is not clear for the reasons that the factors affecting PFV are not only the level of the transportation infrastructure construction, but also the proportion of the secondary industry and the tertiary industry, the population, and the import and export trade of goods. The industrial structure, the population saturation, the pressure of transportation, and even the government's planning and policy guidelines for economic construction may be, therefore, the primary reasons for the slow growth or even the decline in the passenger and freight traffic [51–53]. Thus, the increase in the passenger and freight traffic may be closely related to the improvement of the transportation infrastructure construction. It may not be reasonable to conclude that the transportation infrastructure construction in the region is low based only on the bad calculation results of the SDG indicator 9.1.2 [54].

3. The impact of the transportation infrastructure conditions on the passenger and freight traffic is mainly expressed in whether there are enough roads to supply industries, agriculture, and other departments for the transportation of materials and products or for residents. The calculation results of the SDG indicator 9.1.2 show that the freight volume in Deqing County (especially the road freight volume) has dropped significantly in recent years as if the transportation infrastructure has not had a positive impact on the economy and people's lives. However, this conclusion seems to be inconsistent with the calculation results of the SDG indicator 9.1.1. Road construction has been advancing, while the passenger and cargo traffic has declined yearly. The seemingly contradictory results of these two indicators made us start thinking about the correctness of the result. By site surveying, we found that the main reason for the decline in the passenger and

freight volumes was that Deqing County was seriously affected by its neighboring Hangzhou City. The statistics department of Deqing County only counts the vehicles with local licenses when they compute these two kinds of data, which means if the vehicle working in Deqing County registered its license in Hangzhou city, it will not be counted in the passenger and freight volume in Deqing County. The large number of non-Deqing County license cargo vehicles, therefore, has significantly reduced the amount of freight volumes that can be counted in Deqing County. This method is unreasonable but it is widely used in China, it makes the statistics of Deqing County's freight volume incomplete. The problem will not exist in the national level assessment, but is very common in the study in the county level. Therefore, when applying this indicator in the county level, and combining PFV with TPB to assess the impact of the transport infrastructure construction on the economic development and people's well-being, it may not get the one-sided conclusions from the results of a single indicator and result in a more proper judgment.

4.  The problems of the SDG indicators we found are sourced from the county level research, several types of these problems may also exist in the national-level research. The first problem results from the calculation method of the indicator used. For example, when calculating the RAI indicator, the influence of the unconnected road on the calculation result is ignored, and the distinction between "can find the road" and "can find the road entrance" is not distinguished. The second problem arises because the setting of the indicator has a deficiency, that is, the SDG indicator cannot fully reflect the connotation of the SDG sub-target 9.1. These two kinds of problems may also occur when applying the SDG indicators in other counties and in national level studies, because these problems have little relevance with the field reality in the study area. Moreover, there are problems arising from the real situation in the area. For example, the existence of the urban village makes the urban population have to be taken into account when calculating the RAI indicator. Since urban villages are ubiquitous in most parts in China, this problem may need to be considered also in national level study. In some other countries, there are also urban villages and slums, which are similar to urban villages in China [18], we hope the improved indicators proposed in this paper may provide some useful reference for researches in other parts of the world.

## 6. Conclusions

We proposed an improved indicator system that includes five indicators, namely the modified RAI indicator, passenger and freight volume, road density, accessibility and total postal business in this paper to assess the progress of the SDG sub-target 9.1 in the county level. This study used the data from Deqing County in China to calculate the SDG indicators 9.1.1 and 9.1.2 and the proposed improved indicator system. The following conclusions are obtained by comparing and analyzing the process and results of the indicator calculation:

1.  There are some problems and imperfections in the application of two SDG indicators in the county level. First, the SDG indicator 9.1.1 ignores the difficulties when urban vulnerable groups acquire the transportation infrastructure and the distinction between "search for the road" and "search for the road entrance". In addition, it is often difficult to obtain the urban-rural boundary data needed to calculate this indicator. Second, the SDG indicator 9.1.2 cannot fully embody the impact of the transportation infrastructure construction on people's livelihood. Third, these two indicators cannot fully express the connotation of the SDG sub-target 9.1.

2.  The improved indicator system can eliminate the defects existing in the application of the SDG indicators at the county level. First, the improved RAI index takes into account both the urban and rural populations, and gets rid of the dependence on the urban-rural boundary data. Second, road density and accessibility shows many necessary details of the spatial distribution of the transportation infrastructure construction, which can provide valuable reference information for future development. Third, the passenger and freight volumes and total postal business can be combined to evaluate the impact of the transportation infrastructure construction on the economic

development and people's livelihood. In summary, the improved indicator system can evaluate more accurately, comprehensively, and diversely the level of the transportation infrastructure construction in the county level.

**Author Contributions:** Conceptualization, J.X.; methodology, J.X.; software, J.X.; validation, J.X.; formal analysis, J.X.; investigation, J.X. and J.B.; resources, J.X.; data curation, J.X.; writing—original draft preparation, J.X.; writing—review and editing, J.X. and J.B.; visualization, J.X.; supervision, J.B.; project administration, J.B. and J.C.; funding acquisition, J.B.

**Funding:** This research received no external funding.

**Conflicts of Interest:** The authors declare no conflict of interest.

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
