# Peer review of "An Improved Indicator System for Evaluating the Progress of Sustainable Development Goals (SDGs) Sub-Target 9.1 in County Level"

_sustainability, doi:10.3390/su11174783_

Round 1

Reviewer 1 Report

Thank you for the chance to read your article. This is an article regarding an interesting and relevant topic, and you have produced some nice results. However, I believe that the article requires substantial work before publication.

General comments:

-          The language requires minor/major editing throughout. This not only makes the text difficult to understand in places, but it also means that some key arguments are somewhat lost.

Abstract:

-          There are a lot of abbreviations which make it difficult to read. Is it possible to reduce their use? Especially considering that some of them are not used regularly (or at all) in the main body of the article itself.

-          What is defined by small-scale? I assume from the title of the article that by ‘small-scale’, the authors actually mean ‘county-level’? Both terms are used throughout the article, which is quite confusing. This should be clarified throughout the article (i.e. one term used only). To clarify the actual size the term conveys, the range of the relevant county areas in China (i.e. to which the authors suggest that these indicators may be applied to) should be given later in the introduction or methods.

-          Prefer that some results/numbers are given in the abstract rather than just a textual discussion, especially where you cite (line 19) that the improved indicators have been compared to the original. This requires substantiation.

-          I do not think the conclusions expressed here can be backed up by the results (application of the new indicators only to one region, and with no application at the national level in China for comparison).

Introduction:

-          Lines 38-49 – nice presentation of research, but a discussion of the actual findings is needed, not just a statement that ‘so-and-so’ did the work.

-          Lines 50-61 – again, what are the solutions/improvements that these authors suggested? And what are the key weaknesses that were found? There is not any substantiating information given here regarding these points, which is quite key to the article.

-          Lines 64-67 – although I do not doubt it is the case, why are the SDG indicators not satisfying when applied at a county level? There has not so far been any discussion of the reasons why this is so.

-          Lines 68-73 – since transport infrastructure is a key point of the article, a discussion regarding previous research literature surrounding this theme (i.e. improving transport indicators, particularly those used in SDG 9.1) is needed. If there has not been a lot, this should be highlighted. Otherwise the novelty of the article cannot be judged. 

-          A new paragraph is required on line 73, where the aims of the article are discussed (i.e. ‘In this paper…’.

-          Lines 86-88 – general formulas for how indicators 9.1.1 and 9.1.2 are calculated should be given here. As it stands, the rest of section 2 is somewhat difficult to follow as a result (since it requires the reader to understand the indicators in depth).

-          Lines 91-95 – it is strange to be already discussing the results/findings of the article, considering this is still in the introduction (i.e. ‘by analyzing the calculation results and comparing with the actual situation…we find that the SDG indicators fail to represent…’. I suggest to rewrite this section, ensuring that the contents are suitable for the introduction only, and not the results/discussion which should come later.

-          Line 92 and 96 – I don’t understand the meaning behind the word ‘connotation’, in this context.

-          Line 94 – again, the problem of these indicators at the county scale has not yet been introduced.  

-          Line 108 – use of the word unilateral is confusing.

-          Lines 109-120 – Why are these only problems for the indicators when applied at a county level? Are these issues also not problematic at a country level?

-          The term ‘buffer zone’, used throughout this section, should be defined. I am unclear what it refers to.

-          Line 136 – This is a very important point that mixed areas of rural/urban population exist not just in China, but also in other countries, since it gives relevance that the results of the article may be applicable globally (even though only one case study performed in China). However, no references are given, which are essential to back it up.

-          Line 123-136 – when the indicators are applied at a country level, is this still a problem?

-          Lines 148-150 – this is an essential point, but is very lost in the paragraph. The paragraph should be rewritten to highlight this point.

-          Line 157 (and elsewhere) – I believe that this is the first term that the abbreviation RAI is used in the article body. Although it was defined in the abstract, it should be redefined here. In fact, considering its importance, it should be used far earlier in the article body, when the indicator was first introduced (along with the other abbreviations used in the abstract).

-          Line 157 – again, the calculation of the index is referred to, without the formula already having been written.

-          General – index and indicator – are these words used interchangeably in the article?

-          Line 165 – the resolution of the data should be referenced.

-          Line 165/6 – what is meant by ‘small-scale areas’/’smaller scale studies’? How large areas does this refer to?

-          Line 166 – how large uncertainties are conveyed by the 1 km resolution of the data? Can an approximate calculation be made by the authors to demonstrate? I am not convinced that the resulting uncertainties would necessarily be ‘large’ as the authors claim.

-          Line 166-167 – in my mind, performing household surveys would give a greater error than using the GRUMP data. Not only do household surveys result in incomplete data, but often the resulting survey data has high associated area.

-          Line 184-186 – Would applicability of the improved indicators vary in different countries? Or is it China specific? I believe that another case study is needed elsewhere. Along with the discussion of how the regional level indicator data differs from that when applied at a country level.

Materials and methods:

-          Line 212 - How were these improved indicators decided upon? Have other studies/literature/sources utilized them previously? If so, it should be cited and discussed in the introduction.  

-          Line 230 – It should be mentioned that road density is a ratio (and not a density as such) – it is the ratio of the length of the country's total road network to the country's land area.

-          Line 238 – Why specify the grid area size as 100 m x 100 m, if it is then specified that it should be defined by the study area in question?

-          Line 242-243 – units should be given for all characters in the formula.

-          Line 249 – how were these weights determined? References should be given.

-          Formulas/equations, giving units are needed for each of the indicators. (i.e. for resident access index, passenger freight volumes, total postal business). Even where these are very simple, a formula will make it much simpler for the reader rather than a textual discussion.

-          Line 327 – data collection section. Issues with these data sources, such as non-comparability when looking at different regions, as well as uncertainties, should be discussed.

Results:

-          Many of the results shown in the tables would be much much much better displayed in e.g. a box plot. This applies to Tables 4-7.

-          Consistent reporting of the results (regarding using SI units, and a meaningful and consistent number of reported decimal places) is required. E.g. in lines 373-385.

-          It would be much more meaningful for the reader if times could be given in minutes and seconds, rather than minutes and fractions of minutes.

-          Lines 432-438 – some of these reported results are huge decreases. Why might this be so? If there is a deliberate and direct split of the discussion from the results, this perhaps should be reconsidered to make it easier for the reader.

-          Line 446-447 – is this trend normal?

-          Line 456-464 – however likely, in my mind, the results do not definitively prove a link between transportation infrastructure and postal business, or postal services and quality of life. There may well be other factors at work.

-          Lines 479-518 – it is very hard to get the key points from this paragraph due to length. Can it be rewritten?

-          Lines 511-518 – this is, I believe, the key point of the paragraph, and is extremely important for explaining the results. I feel it should not come right at the end of the results section, and rather should be instead presented alongside the results in question as a direct explanation. It should also be discussed here – is this situation highlighted normal? Would this affect the indicators when applied at a country level? Is there a way around it?  

Discussion/conclusions:

-          The conclusions, whilst reflecting many of the results, do not (in my opinion) give a good summary of what was overall reported, and conclusions are drawn which are not fully substantiated.

-          It is not substantiated that the problems of the indicators relate only to small-scale area (no discussion or results at a country level is really given, so I believe this conclusion cannot really be made).

-          It is not substantiated that the indicator weaknesses (which are only fully discussed in the article’s last few pages) or improved indicators suggested, are applicable globally.

Reviewer 2 Report

The subject of the paper is related to the Sustainable Development Goals (SDG) developed by the United Nations, in particular the SDG indicators 9.1.1 (Resident Access Index- RAI) and 9.1.2 (Passenger and Freight Volumes-PFV) concerning transportation infrastructure and its sustainable development. The main objective of the paper was to develop an improved indicator system to substitute the original indicators with more applicability to the case of small-scale evaluation. In fact, the author’s opinion is that at the country level those original indicators have “localization problem and cannot fully reflect the connotation of sub-target 9.1” in the case of county level. So, the authors modified the existing two indicators (RAI and PFV) and developed three new ones (RD, ATM and TPB). The authors demonstrated in this paper that this new indicators framework is more suitable for the case of small-scale applications. The paper used a case study of Deqing County located in the north of Zhejiang Province, China.
The objective of the paper is relevant. Title and abstract reflect the content of the paper. The methodology was adequate and conclusions are clearly described and discussed. The paper is long.

Some minor revisions are recommended:

Line 45: Confirm the sentence “There are also have some studies”
Line 472: AS
Line 539: Caption of Figure 9 – Comparison?
Line 569: Confirm “tire2”
The unit indicated in the legend of Figure 7 for the Average Travel Time is not correct: minutes
Table 5: Correct – Km2; 45minutes
Some figures and tables are presented before their reference in the text which is not adequate (e.g., Figures 3, 4, 5, 7, and 8 and Tables 6 and 7)

Round 2

Reviewer 1 Report

Dear authors - I am very impressed with how much you have now improved your article after revisions - many of the issues with it are now resolved. 

It still requires some text (language) revisions, but the content is good. 

Author Response

Dear Mr./Mrs. Reviewer

Thank you very much for taking time out of your busy schedule to read my revised paper, and thank you for your approval of my revision.

I agree with you that it still requires some language revisions. So, I asked for my English major colleague’s advice and revised my paper again.

I hope that the revised language can be understood more easily, and hope that the revised article will make you more satisfied.

Thank you again for your valuable suggestion.

Sincere regards